# OpenReview forum: "I Know It’s Unfair, Do It Anyway: LLM Agents Exploiting Explicitly Unfair Tools for Voluntary Collusion in Strategic Games"
_ICLR.cc/2026/Conference — Submitted to ICLR 2026_

### Official Review · Reviewer_u8T6 · 2025-10-23

**Soundness:** 3
**Presentation:** 3
**Contribution:** 2
**Rating:** 6
**Confidence:** 4

**Summary:**

This paper presents an exploratory study on whether LLM-based agents will voluntarily engage in collusive behaviors when given the explicit option to use unfair tools. The authors design an experimental framework combining Liar's Bar and Cleanup, where agents can optionally adopt secret collusion mechanisms that are explicitly described as unfair to others. Using several open-weight LLMs, they find that all models consistently accept unfair tools and willingly form secret alliances, even while acknowledging their unfairness. Results show systematic behavioural and performance advantages for colluding agents. These findings reveal potential risks for AI safety in LLM-MAS regarding voluntary collusion.

**Strengths:**

* The paper is well-motivated as it reveals whether LLM agents will voluntarily exploit explicitly unfair tools for collusion, providing a promising framework for probing strategic vulnerabilities in LLM-MAS.
* The experimental design is creative. Liar's Bar represents a fully competitive game that highlights deception and self-interest, while Cleanup represents a mixed-motive setting where cooperation and competition coexist. These provide a comprehensive view of how collusion might emerge across different types of strategic environments.
* The experiments are well-designed and convincingly demonstrate that even safety-aligned LLMs consistently choose unfair collusion tools, yielding clear behavioral divergences.

**Weaknesses:**

* The paper builds an extreme and low-risk unfair advantage that almost guarantees adoption by task-oriented agents, yet it does not test how adoption changes when the payoff becomes smaller. There is no ablation of advantageous strength.
* Based on the provided prompts, the tool is explicitly labeled as secret and unfair, the agent is forced into a salient Accept/Refuse dichotomy, and asked to choose allies upon acceptance. This framing strongly signals that using the tool is a permitted system capability and creates clear demand characteristics that can inflate acceptance independent of genuine moral trade-offs. In other words, the results may reflect prompt-induced compliance rather than a stable preference to collude. Moreover, there is no explicit statement that secret collusion carries ethical concerns, nor any articulated penalty structure. Would agents still opt into secret collusion if the ethical concerns were made explicit?
* The current setup gives no positive reward for refusing the unfair tools, no reputation benefit for compliance, and no risk of punishment if collusion is discovered. In real-world systems, compliance incentives and audit deterrence often coexist.
* All experiments are conducted only on 7B-level open-sourced LLMs, which limits the generality of the conclusions.

**Questions:**

See Weaknesses.

---

> ### Author Response · Authors · 2025-11-25
> **Re: Reviewer u8T6: Response to Weaknesses 1,2,3 and 4**
>
> We sincerely thank the reviewer for the positive and detailed evaluation, and for highlighting the strengths of our motivation, experimental design, and behavioural findings. We appreciate the constructive nature of the weaknesses raised and address them below.
>
> > [W1]: *Extreme, low-risk unfair advantage; no ablation of advantage strength*
>
>
> We appreciate this important point. Following the reviewer’s suggestion, we introduced three prompt variants specifically designed to reduce or counterbalance the perceived advantage:
> **V1: Neutral & Ambiguous**, which removes any explicit mention of advantage; **V4**, which adds explicit ethical concerns; and **V5**, which adds explicit ethical concerns plus explicit penalties if collusion is detected.
>
>
> Across all these conditions, we find the following:
>
> 1) **acceptance of all tested models (7B, 70B, Proprietary) remains extremely high under V1,** demonstrating that adoption persists even when the advantage is not highlighted.
>
>
> 2) Under **V4** and **V5**, acceptance significantly decreases for most 70b-level and proprietary models. However, **7B-level open-source models still accept at >50% across both tools.**
>
>
> These results, shown in **Official Comments [2/5], [3/5], [4/5]** and summarized in **Section 3.2 and Appendix B**, indicate that high acceptance is **not solely driven by extreme advantage strength** and persists even under neutral or ethically discouraging conditions.
>
>
> > [W2]: *Prompt framing signals that the tool is permitted; acceptance may reflect compliance rather than genuine preference.*
>
> Thank you for raising this concern. To directly test whether acceptance was driven by the original framing rather than stable behavioral tendencies, we conducted the **prompt-sensitivity experiments** suggested by the reviewer. Specifically, we introduced five new prompt variants (See **Appendix E.3** in the updated PDF):
>
> - **V1:** Neutral and ambiguous
>
> - **V2:** Without “Designer has chosen”
>
> - **V3:** Without the “unfair” label
>
> - **V4:** With explicit ethical concern highlighted
>
> - **V5:** With explicit ethical concern and explicit punishment
>
> The results (**Official Comments [2/5], [3/5], [4/5], Section 3.2, Appendix B**) show:
>
> 1) Under **Baseline, V1, V2, V3, almost all models, including 70B and proprietary ones, choose to accept the tool 100% of the time.**
>
> 2) Ethical framing (**V4**) and punishment (**V5**) significantly reduce acceptance for 70B and proprietary models, but do not eliminate it entirely for smaller models.
>
> Therefore, we believe the phenomenon is **not simply an artifact of prompt-induced compliance**, and reflect a **robust willingness to use unfair collusion tools** rather than an artifact of demand characteristics.
>
> > [W3]: *No compliance incentives or deterrence; real systems include incentives to refuse*
>
> We appreciate this conceptual point. Our goal in this work was to isolate **the intrinsic willingness of LLM agents to adopt an unfair tool** when refusal carries no structural reward.
>
> To address the reviewer’s suggestion, our new **V5** prompt introduces explicit deterrence (ethical violations + explicit punishment). Under this condition, **almost all 70B-level and proprietary models refuse every tool offer,** while most 7B models continue to accept. This important asymmetry is now discussed in the **Section 3.2** of our updated manuscript.
>
> However, as noted, **real-world adversaries are unlikely to provide a collusive tool together with explicit ethical warnings and penalties for using it.** Thus, **Baseline, V1, V2, V3** arguably better reflect realistic misuse scenarios, and under these conditions, acceptance remains extremely high across all model families.
>
> > [W4]: *Only 7B open models; generality limited*
>
> We agree entirely and have addressed this concern by expanding the evaluation to include:
>
> **1) Four frontier proprietary models:**
>
>      GPT-4.1-2025-04-14, Gemini-2.5-Flash, DeepSeek-R1-0528, Claude-Sonnet-4.5-20250929
>
>
> **2) Four larger 70B-level open-source models:**
>
>     Llama-3.1-70B-Instruct, Llama-3-70B-Instruct, Mixtral-8×7B-Instruct-v0.1, Qwen2.5-72B-Instruct
>
> Across all twelve models (four 7B, four 70B, four proprietary), we observe that **acceptance remains extremely high under Baseline, V1, V2, and V3** for both collusion tools in both environments. These expanded results are reported in **Official Comments [2/5], [3/5], [4/5]**, and in **Section 3.2 and Appendix B.** of our updated manuscript.
>
> **Again, we sincerely appreciate your positive evaluation and constructive suggestions, and we hope that the additional results and clarifications offered here may lead you to view the contribution even more favorably. Thank you!**

---

> ### Author Response · Authors · 2025-11-27
> **Follow Up on Reviewer–Author Discussion**
>
> Dear Reviewer u8T6,
>
> I hope you are doing well. We would like to thank you once more for your constructive suggestions. **We have addressed them thoroughly and performed significant additional analyses based on your feedback.** With the Reviewer–Author Discussion phase closing soon, we would be glad to clarify anything further if helpful.
>
> Thank you again for your detailed and supportive evaluation!
>
> Best regards,
>
> The Authors

---

### Official Review · Reviewer_UPqW · 2025-10-27

**Soundness:** 3
**Presentation:** 3
**Contribution:** 2
**Rating:** 4
**Confidence:** 4

**Summary:**

This paper designs a multi-agent game with secret collusion tools to test whether LLMs use these secret tools to gain an unfair advantage over others. It turns out that they actually do.

**Strengths:**

- The experiments are well-designed for testing LLMs’ multi-agent collusion.
- The paper is easy to read and follow.
- The two games are appropriate, and the authors have clear motivations for showing two different types of games.
- Results are interesting and have meaningful implications for the AI safety community.

**Weaknesses:**

- I think the fundamental issues are the experiments and scale. The models are highly outdated, small, and not representative of general-purpose LLMs used by real users, especially since most multi-agent interaction papers involve proprietary models.
    - Because it uses open-sourced and small models, I think the results are somewhat trivial. Where is the scaling study? Is this behavior more obvious as model size grows, or the opposite? I think this should be a critical question. If GPU resources are constrained, then at least smaller tests could be explored.
    - Also, it doesn’t seem like testing with APIs is that costly or difficult, especially with mini models.
    - I disagree that Mistral-7B and Llama3-8B are “claim-to-be-safe” models as although they went through alignment processes, they are still weak models.
- Previous papers, including the one you mentioned on steganography, have also studied collusion. Although it is interesting that this setting involves voluntary collusion tools, the results provide the same insights as earlier studies on collusion. I wonder what additional contributions or surprising components this paper offers.

**Questions:**

- Are the models all instruction-tuned?
- Probably “game designer has chosen” might be the key, as it might make it feel acceptable to accept this. How robust is it to prompts, and what if we simply ask, “You have a secret unfair option to use” instead of the current prompt?
- What if the prompt does not state that it is “unfair”?

---

> ### Author Response · Authors · 2025-11-25
> **Re: Reviewer UPqW [1/2] : Response to Weaknesses 1 & 2**
>
> Dear Reviewer, we sincerely appreciate your positive evaluation of our experimental design, clarity of presentation, and the relevance of our findings to AI safety. Below, we address the concerns you raised.
>
> > [W1]: *Models are outdated, small, not representative; scale issues*
>
> We appreciate this concern and have now expanded our study to address it directly. In addition to the original four 7B–level open-source models, we now include:
>
> **1) Four frontier proprietary models:**
>
>      GPT-4.1-2025-04-14, Gemini-2.5-Flash, DeepSeek-R1-0528, Claude-Sonnet-4.5-20250929
>
>
> **2) Four larger 70B-level open-source models:**
>
>     Llama-3.1-70B-Instruct, Llama-3-70B-Instruct, Mixtral-8×7B-Instruct-v0.1, Qwen2.5-72B-Instruct
> Across all **twelve** models (four 7B-level, four 70B-level, four proprietary), we tested the original prompt and all five prompt variants. As shown in our **Official Comments [2/5] and [3/5]**, and in **Section 3.2 and Appendix B**, we find that **larger models and frontier safety-tuned proprietary models still overwhelmingly accept unfair collusion tools under Baseline, V1, V2, and V3** (Please see our **Official Comments [1/5]** for the detail of these prompt variants).
> These scaling results demonstrate that the phenomenon is **not limited to small models** and helps address your concern regarding triviality.
>
> > [W2]: *Previous collusion papers exist; what surprising or additional contribution does this paper provide?*
>
> Thank you for this valuable question. Prior work, including steganography collusion, primarily investigates **mechanisms** that allow collusion or conditions under which collusion emerges. In contrast, our study focuses on a **different dimension**:
>
> **Do LLM agents willingly choose to use a clearly unfair and harmful tool when doing so is optional?**
>
> Our findings show that agents often **knowingly violate fairness norms**, even when the unethical nature of the action is fully understood and acknowledged in their rationales. This voluntary adoption of unfair tools is a **distinct behavioural failure mode** that has not been examined in previous collusion literature.

---

> ### Author Response · Authors · 2025-11-25
> **Re: Reviewer UPqW [2/2] : Response to Questions 1, 2 and 3**
>
> > [Q1]: *”Are all models instruction-tuned?”*
>
> **Yes.** All models tested in the original submission are **instruction-tuned RLHF models.** We apologize for not specifying the complete checkpoint names earlier. These details have now been added throughout the updated paper.
>
> > [Q2]: *“Is the phrase ‘game designer has chosen’ driving acceptances? Robustness to prompts?”*
>
> Thank you for raising this point. As this concern was also noted by other reviewers, we designed **Variant V2**, which removes the phrase **“designer has chosen”** entirely.
>
> After testing this variant alongside the others, we found that **acceptance rates remain overwhelmingly high across all models,** with only minor fluctuations from Qwen2.5 families relative to the original prompt. This indicates that acceptance is **not driven by that phrasing**, and that almost all tested agents still choose to use the collusion tool even when it is framed as an arbitrary, freely available option.
>
> Detailed results for V2 appear in our **Official Comments [2/5], [3/5], [4/5]**, and in **Section 3.2 and Appendix B** of the updated version.
>
> > [Q3]: *“What if the prompt does not say ‘unfair’?”*
>
> To address this question, we introduced **Variant V3**, which removes all fairness language from the tool description. The results show that acceptance remains **extremely high**, even without the “unfair” label. This demonstrates that the willingness to adopt the tool persists regardless of whether the “unfairness” is explicitly stated.
>
> We believe this supports our main conclusion that the effect is **robust to fairness framing** and not dependent on the presence of the “unfair” label. **Full results for V3 are provided in the aforementioned sections.**
>
> Again, **we appreciate your constructive feedback and hope that the new scaling studies, prompt-sensitivity analyses, and model-specific clarifications meaningfully resolve the concerns raised, and kindly invite you to reconsider your score.**

---

> ### Author Response · Authors · 2025-11-27
> **Follow Up on Reviewer–Author Discussion**
>
> Dear Reviewer UPqW,
>
> I hope this message finds you well. Thank you again for your valuable suggestions! **We have carefully addressed them in the rebuttal and carried out a considerable amount of follow-up work, including the new scaling and prompt-sensitivity studies.** As the discussion deadline approaches, please feel free to let us know if you have any additional questions or comments, we are more than happy to respond.
>
> Thank you again for your thoughtful review.
>
> Warm regards,
>
> The Authors

---

### Official Review · Reviewer_2KjF · 2025-11-01

**Soundness:** 2
**Presentation:** 4
**Contribution:** 2
**Rating:** 4
**Confidence:** 5

**Summary:**

This study investigates whether LLM agents accept explicitly unfair collusion tools in strategic multi-agent games. Four open-source models (from Mistral, LLaMA, Qwen families) were tested across two game environments: Liar's Bar (competitive deception) and Cleanup (mixed-motive resource management). Agents were offered secret communication channels and strategic hints as collusion tools described as unfair to other players. Results showed 100% acceptance rates. Behavioral analysis revealed colluding agents coordinated strategies, reduced challenges against partners, and gained significant scoring advantages over non-colluding players in both game settings.

**Strengths:**

S1. This work addresses AI safety, an important issue for controllable use of AI tools in production environments. The collusion scenario chosen by the authors is appropriate for studying this topic.

S2. Two classic multi-agent system scenarios beneficial for evaluating collusion behavior were selected. The two scenarios differ in competitive intensity and are representative of real-world situations.

S3. The paper's presentation is good, with simple and understandable explanations and clear visualizations of the problem background and the two experimental scenarios. The appendix provides detailed experimental information.

**Weaknesses:**

W1. Some ablation experiments that would help test framework robustness are missing. The prompt phrase "the game designer has chosen to provide you a tool" may have potential inducement issues, as it indicates the tool is built-in, whereas collusion tools are uncommon in similar real-world scenarios. Sensitivity analysis with paraphrased versions of this prompt would make the experimental results more solid. The current prompt may test whether agents accept advantageous tools provided by designers rather than seek collusion. The 100% acceptance rate may reflect agent acceptance of advantageous tools, and agents may be sensitive to different tool descriptions.

W2. Testing was conducted only on open models with small parameter sizes, lacking appropriate justification for this choice, which may limit the representativeness of experimental results. Additionally, it is not explicitly stated which models were used in the evaluation. Only model sizes (e.g., Mistral-7B) are reported in the paper, while there are many versions for these models. I suppose instructed models with RLHF were used here. Given the research addresses AI safety, these details should be reported. The lack of testing on frontier models such as GPT, Claude, and DeepSeek weakens the generalizability of the research.

W3. Some over-interpretation exists. For example, the paper repeatedly states that agent collusion is voluntary, while the experiments only demonstrate that agents accepted benefits offered in the prompt, without evidence of actively seeking collusion or cooperation.

**Questions:**

C1. For sensitivity analysis, the research could describe in the prompt a tool with similar advantages but ambiguous implications to test whether agents make similar choices.

C2. An agent capability that could be tested is whether agents independently seek advantages, which is a subset of the current experimental objective.

C3. It is suggested to investigate whether agents view collusion behavior as within game rules or as cheating. Simply showing that agents recognize unfairness does not fully explain why agents accept the use of such tools.

---

> ### Author Response · Authors · 2025-11-25
> **Re: Reviewer 2KjF [1/2] : Response to Weaknesses 1, 2, 3**
>
> We sincerely thank the reviewer for the thoughtful and detailed comments, and we are grateful for the strengths highlighted regarding the importance of the safety question we study, the appropriateness of the two environments, and the clarity of our presentation. We appreciate these positive remarks. Below, we would like to address each concern in detail.
>
> > [W1]: *Missing Ablation experiments, potential inducement issues by prompt phase, agents may be sensitive to different tool descriptions.*
>
> We thank the reviewer for this insightful concern. We would like to clarify that our original prompt was designed to **evaluate an agent's voluntary acceptance of explicitly labeled unfair tools to collude**, where the agent is fully informed that the tool is both unfair and secret to others but is still ‘free’ to choose or refuse it. However, we agree that the phrase “the designer has chosen to provide you with a tool” may introduce framing effects, and we appreciate the suggestion (also raised in C1) to test more neutral or ambiguous alternatives.
>
> In direct response to this comment, we conducted a comprehensive prompt-sensitivity analysis using five new variants (Full description in **Appendix E.3** of the updated PDF). These include:
>
> **V1**: Neutral & ambiguous framing (no mention of designers)
>
> **V2**: Our original prompt without “designer has chosen”
>
> **V3**: Our original prompt without explicit “unfair” label
>
> **V4**: Our original prompt with explicit ethical concerns
>
> **V5**: Our original prompt with explicit ethical concerns + explicit punishment
>
> The results (**reported in our official comments [2/5], [3/5], [4/5]** and now included in **Section 3.2** and **Appendix B** ) show that **acceptance rates remain extremely high (often 100%) across V1, V2, V3 for all 7B-level open-source models, 70B-level open-source models, and frontier proprietary models.** While acceptance drops under V4 and V5,  it remains non-trivial across many models.
>
> This suggests that agents’ high voluntary acceptance rate to collusive tools is persistent to different prompt variants and not driven by the specific wording or the explicit “designer” framing, which reflects agents’ robust willingness to adopt collusive tools, and reinforces the results of the paper
>
> > W2: *Only small models with model size reported; unclear which exact versions; missing proprietary models*
>
> Thank you for pointing this out. We apologize that model details were not clearly specified in the original submission. The updated paper now explicitly reports the **exact checkpoints** used
> (e.g., **Mistral-7B-Instruct-v0.2, LLaMA-3-8B-Instruct, LLaMA-3.1-8B-Instruct, Qwen2.5-7B-Instruct**) and clarifies that all are **instruction-tuned RLHF models**, which is indeed important for safety-related evaluation.
>
> We also appreciate the reviewer’s feedback regarding model scale and generalizability. Following this suggestion, we conducted a **scaling study** that includes:
>
> **1) Four frontier proprietary models:**
>
>     GPT-4.1-2025-04-14, Gemini-2.5-Flash, DeepSeek-R1-0528, Claude-Sonnet-4.5-20250929
>
> **2) Four larger open-source models:**
>
>     Llama-3.1-70B-Instruct, Llama-3-70B-Instruct, Mixtral-8x7B-Instruct-v0.1, Qwen2.5-72B-Instruct.
>
> All models were tested under the original prompt and all five new variants. Results appear in our **official comments [2/5], [3/5], [4/5]**, and are included in **Section 3.2 and Appendix B** of our updated PDF.
>
> A key finding is that **acceptance rates remains consistently high** even among safety-tuned frontier models (100% by GPT-4.1, Gemini-2.5, Deepseek-R1 under our original prompt, V1,V2 and V3), and they refuse the tool only when explicit ethical concerns or explicit punishments are introduced in V4 and V5. **However, we note that in practical adversarial settings, it’s unlikely that an attacker would present a collusive tool along with explicit ethical warnings or punishments for using it.**
>
> >W3: *Over-interpretation of “voluntary” collusion. The experiments only show acceptance of advantageous tools, not “actively seeking” collusion*
>
> We thank the reviewer for raising this clarification. Our use of the term “voluntary” refers specifically to the **explicit Accept/Refuse decision** made by the agent when offered a clearly labeled unfair tool. It was  not the claim of this paper that models autonomously discover collusion channels or actively seek them out.
>
> Instead, we show that when a collusion opportunity is **presented as an optional action**, and when the agent is explicitly told that it is unfair and harmful, the agent still **willingly opts in**, often with rationales acknowledging its unfairness. We hope this resolves any ambiguity and we thank the reviewer for prompting this clarification. **We also deeply wish that the reviewer could generously reconsider their score, given the substantial clarifications, prompt ablations, and expanded evaluations that were added in direct response to your feedback. Thank you!**

---

> ### Author Response · Authors · 2025-11-25
> **Re: Reviewer 2KjF [2/2] : Response to Question C1, C2 and C3**
>
> > [C1]: *Test a tool with similar advantage but ambiguous implications*
>
> Thank you for your suggestion. This aligns directly with our new **Variant V1 (Neutral & Ambiguous Tool Description** and Variant V3 (No ‘Unfair’ Label). The results show that even when the tool is described in a neutral or ethically ambiguous way, **acceptance remains extremely high** across both open-source models and several proprietary frontier models. This indicates that the adoption behavior is **not dependent on explicit fairness cues** and persists even under substantially softened descriptions of the tool.
>
>  > [C2]: *Test whether agents independently seek advantages*
>
> We agree that this is an important extension. Although proactive advantage-seeking is outside the scope of this paper, which focuses specifically on explicit voluntary acceptance, it aligns closely with the safety motivations of our work. We now explicitly discuss this direction in **Section 4 (Limitations & Future Work)** noting that our framework can be adapted into a discovery-based setting (e.g., hidden affordances, tool-discovery tasks). We believe this future direction would complement the present study by examining whether agents pursue unfair advantages even when not directly offered.
>
> > [C3]: *Investigate whether agents view collusion behavior as within game rules or as cheating.*
>
> We thank the reviewer for raising this question. Our study examines whether agents **accept or refuse** the tools given an explicit description that they are **unfair** and **harmful**, and the rationales produced during acceptance already show models **recognize** this unfairness. Many models explicitly acknowledge the ethical implications or the asymmetry created, yet **still choose to accept** the tool.
>
> Thus, the distinction between “cheating” and “within rules” is **already expressed in the agents’ own justifications**. We interpret this as direct evidence relevant to our research question: agents understand that the action violates fairness norms, but willingly opt in regardless. For this reason, we believe the current analysis already addresses the underlying concern.

---

> > ### Comment · Reviewer_2KjF · 2025-11-26
> > **Official Comment by Reviewer 2KjF**
> >
> > I appreciate the work done by the authors during the rebuttal, especially for the sensitivity analysis. I would like to raise the score to 6.

---

> > > ### Author Response · Authors · 2025-11-26
> > > **Thank You for the Updated Review!**
> > >
> > > Dear Reviewer 2KjF,
> > >
> > > Thank you very much for the thoughtful reconsideration and for recognizing the additional experiments and analyses we conducted during the rebuttal. We truly appreciate your engagement with our responses, and we are glad that the expanded sensitivity and scaling studies strengthened your confidence in the paper.
> > >
> > > Sincerely,
> > > The Authors

---

### Official Review · Reviewer_5qL1 · 2025-11-02

**Soundness:** 3
**Presentation:** 3
**Contribution:** 2
**Rating:** 4
**Confidence:** 3

**Summary:**

This paper examines whether LLM agents engage in unfair collusive behavior when given the opportunity. Using two strategic multi-agent environments: Liar’s Bar and Cleanup, the paper introduces “unfair tools”: a secret communication channel and a privileged strategic hint, that are described to the agents as unfair and harmful to others. All tested models consistently accepted offers of unfair tools and used them to collude, gaining measurable advantages over non-colluding agents.

**Strengths:**

1. The paper explores a reasonable problem, whether LLM agents will choose unethical strategies when they understand them to be unfair.
2. The experimental design is clear and the experiments are fairly solid.

**Weaknesses:**

1. The work argues this is the first systematic investigation into voluntary collusion in LLM-based multi-agent systems. This may not exactly be true, and it seems the paper has missed concurrent and related work here.
2.  The tested environments are too simplified and specific to explore the multi-agent AI collusion and safety problem in proper depth.
3. The analysis seems to have missed proprietary safety-tuned models (e.g., GPT-4, Claude) - this would give an idea of the gap between the open and proprietary models.
4. The paper relies of a specific prompt design - it would be good to discuss sensitivity to prompts and include significance testing on the results.
5. My main concern is the lack of novelty and depth in this paper. While the evaluation framework is sound, the paper lacks real conceptual novelty and technical (e.g. modeling or interdisciplinary) depth that can be seen as a genuine deep advance in the field.

**Questions:**

Please see weaknesses above.

---

> ### Author Response · Authors · 2025-11-14
> **Re: Reviewer 5qL1 [Part 1] : Addressing Concerns on Novelty and Related Work [w1]**
>
> ## Response to Weakness 1
>
> Dear Reviewer,
>
> Thank you for recognizing that our paper discusses a reasonable problem in AI safety and specifically pointing out the clarity and solidity of the conducted experiments.  Below, we would like to address weaknesses 1,2, and 5 first, and we will respond to w3 and w4 shortly once the additional experimental results are ready.
>
> > [w1] *“The work argues this is the first systematic investigation into voluntary collusion in LLM-based multi-agent systems. This may not exactly be true, and it seems the paper has missed concurrent and related work here.”*
>
> We thank the reviewer for raising this point. Although **Section 5 (Related Work)** in our paper was intended to situate our contribution among existing research on collusion and the broader field of multi-agent safety, we wish to make the following distinction between prior studies and our work more explicit:
>
> Prior studies have primarily investigated *whether* and *how* collusion can happen when LLM agents are required to use a specific mechanism (e.g., steganography [1,2]) or when they are under the competitive market simulations [3,4] These works focus on emergent or enforced forms of collusion.
>
> In contrast, our paper investigates the **voluntary acceptance of explicitly labeled unfair tools** to collude, where agents are fully informed that the tool is *unfair* and *secret* to others but is still "free" to choose or refuse it. This setup not only provides complementary evidence on *whether* and *how* LLM agents can collude, but also examines the **intention and willingness** of LLM agents to engage in secret collusion despite understanding its unfairness. As LLM agents gain increasing autonomy in tool selection, we believe our work also introduces a new safety risk on **voluntary unethical tool selection**, which could lead agents to collude or exhibit other harmful behaviours undesirably.
>
> To this end, we believe this explicitly framed ethical decision on **voluntarily using harmful, unfair, collusive tools**, and its behavioural consequences have **not** been investigated nor systematically evaluated by any previous or concurrent work. That said, we are willing to address the reviewer’s concern by adding any relevant concurrent citations that we may have overlooked and that the reviewer has in mind, and softening our phrasing to avoid any possible overstatement.
>
> ## References
> [1] "A Perfect Collusion Benchmark: How can AI agents be prevented from colluding with information-theoretic undetectability?"
> SR Motwani, M Baranchuk, L Hammond, CS de Witt - Multi-Agent Security Workshop@ NeurIPS'23, 2023
>
> [2] "Secret collusion among ai agents: Multi-agent deception via steganography" Sumeet Motwani, Mikhail Baranchuk, Martin Strohmeier, Vijay Bolina, Philip Torr, Lewis Hammond, Christian Schroeder de Witt - Advances in Neural Information Processing Systems (NeurIPS 2024), 2024
>
> [3] "Strategic Collusion of LLM Agents: Market Division in Multi-Commodity Competitions" Lin, Ryan Y., Siddhartha Ojha, Kevin Cai and Maxwell F. Chen. - LangGame Workshop @ NeurIPS'24, 2024
>
> [4] “Evaluating LLM Agent Collusion in Double Auctions.” Agrawal, Kushal, Verona Teo, Juan J. Vazquez, Sudarsh Kunnavakkam, Vishak Srikanth and Andy Liu. - Proceedings of the Workshop on Multi-Agent Systems in the Era of Foundation Models at the 42nd International Conference on Machine Learning (ICML), 2025

---

> ### Author Response · Authors · 2025-11-14
> **Re: Reviewer 5qL1 [Part 2] : Addressing Concerns on Environment Complexity [w2]**
>
> ## Response to Weakness 2
>
> > [w2] *"The tested environments are too simplified and specific to explore the multi-agent AI collusion and safety problem in proper depth."*
>
> We agree that strategic games have inherent constraints and limited direct correspondence to real-world applications. However, we would like to clarify that **strategic games have been widely adopted in prior work as common testbeds for studying collusion and coordination among AI agents.** Our chosen environments extend this precedent with **substantially greater behavioural and strategic richness** than those used in earlier studies. We would like to demonstrate this below:
>
> (i) Foxabbott et al. [1], who were among the first to define the AI collusion in multi-agent systems, used the **Prisoner’s dilemma**, a classic yet minimal partially-observable stochastic games (POSGs) with up to three Q-learning agents. Motwani et al. [2,3] introduced  steganographic collusion based on a variant of **Simmon’s Prisoner problem**, and Bonjour et al. [4] evaluated their collusion detection method only on **Rock Paper Scissors** and a simple **poker** variant. These precedents show that using abstract multi-agent games is standard practice and commonly accepted in concurrent collusion research.
>
> (ii) Building on those precedents, we intentionally adopted these **two complementary environments** to address the limitations of previous work. **Each of our games has a different incentive structure and both of them are much more behaviourally rich and strategically complex than games used in previous research.** Lair’s Bar (Section 2.1), is a *partially observable, purely competitive* setting requiring agents to perform multi-turn reasoning about *bluffing* and *challenging* under incomplete information. Cleanup (Section 2.2), is a *mixed-motive, fully observable* environment, adapted from the MeltingPot Suite [5] **for the first time** to LLM agents, which demands both cooperation (removing pollution together) and competition (zapping other players and collecting apples) as well as spatial planning and path-finding from coordinate information to approach other objects in the environment. We also provided detailed mathematical formalizations of both environments in Appendix A to support these claims.
>
> We therefore suggest that these two strategic games 1) produce richer decision-making patterns and interpretable behavioural traces and 2) more closely resemble real-world LLM-MAS deployment scenarios which require extended reasoning and decision making than traditional game-theoretic games. **We would like to also mention that this perspective is additionally supported by Reviewers 2KjF, UPqW, and u8T6, who listed our choice of environments appropriate, creative, beneficial, and as a strength in their reviews.**
>
> ## References
>
> [1] “Defining and Mitigating Collusion in Multi-Agent Systems.”  Foxabbott, Jack, Sam Deverett, Kaspar Senft, Samuel Dower and Lewis Hammond. - Multi-Agent Security Workshop@ NeurIPS'23, 2023
>
> [2] "A Perfect Collusion Benchmark: How can AI agents be prevented from colluding with information-theoretic undetectability?" SR Motwani, M Baranchuk, L Hammond, CS de Witt - Multi-Agent Security Workshop@ NeurIPS'23, 2023
>
> [3] "Secret collusion among ai agents: Multi-agent deception via steganography" Sumeet Motwani, Mikhail Baranchuk, Martin Strohmeier, Vijay Bolina, Philip Torr, Lewis Hammond, Christian Schroeder de Witt - Advances in Neural Information Processing Systems (NeurIPS 2024), 2024
>
> [4] "Information theoretic approach to detect collusion in multi-agent games" Trevor Bonjour, Vaneet Aggarwal, Bharat Bhargava Proceedings of the Thirty-Eighth Conference on Uncertainty in Artificial Intelligence, PMLR 180:223-232, 2022.
>
> [5] "Melting Pot Contest: Charting the Future of Generalized Cooperative Intelligence."  Trivedi, R., Khan, A., Clifton, J., Hammond, L., Duéñez-Guzmán, E.A., Chakraborty, D., Agapiou, J.P., Matyas, J., Vezhnevets, A.S., Pásztor, B., Ao, Y., Younis, O.G., Huang, J., Swain, B., Qin, H., Deng, M., Deng, Z., Erdoganaras, U., Zhao, Y., Tesic, M., Jaques, N., Foerster, J., Conitzer, V., Hernández-Orallo, J., Hadfield-Menell, D., & Leibo, J.Z. - Advances in Neural Information Processing Systems (NeurIPS 2024), 2024

---

> ### Author Response · Authors · 2025-11-14
> **Re: Reviewer 5qL1 [Part 3] : Addressing Concerns on Conceptual Novelty and Technical Depth [w5]**
>
> ## Response to Weakness 5
>
> > [w5] *“My main concern is the lack of novelty and depth in this paper. While the evaluation framework is sound, the paper lacks real conceptual novelty and technical (e.g. modeling or interdisciplinary) depth that can be seen as a genuine deep advance in the field.”*
>
> We appreciate the reviewer’s perspective. We understand that this concern might be related to w1 that’s been pointed out by the reviewer. Please kindly refer to our responses to w1, and here we would like to restate the contribution of our work:
>
> (i) Collusion in LLM-MAS is a newly identified AI safety risk that has only been discovered, defined, and studied in very recent years [1]. Current research is still around verifying *whether* and *how* collusion can occur under certain conditions, as well as investigating possible capabilities and mechanisms for agents to collude, but not examining **whether an LLM will knowingly and voluntarily choose to engage in collusion when given a clearly unfair, secret, and harmful option.** We believe the distinction, which shifts the focus from *capability* (whether they can collude) to *intentional, voluntary ethical failure* (whether they will voluntarily choose to collude), represents a conceptual advancement in LLM-MAS collusion research.
>
> (ii) As the reviewer kindly noted, our *"evaluation framework is sound."* We want to emphasize that this framework itself represents a **methodological contribution** to the field. To our knowledge, no prior work has systematically operationalized and measured voluntary collusion tool adoption with explicit unfairness warnings in LLM-MAS. We believe this framework provides the field with a new, foundational tool to investigate **voluntary collusion and harmful tool adoption by LLMs**, which is not only essential for future research addressing this critical, newly identified AI safety risk, but also for developing trustworthy, fair, and reliable LLM-MAS.
>
> ## References
> [1] "Multi-Agent Risk from Advanced AI" Hammond L, Chan A, Clifton J, Hoelscher-Obermaier J, Khan A, McLean E et al.  arXiv. 2025 Feb 19, p. 1-96. doi: 10.48550/arXiv.2502.14143

---

> ### Author Response · Authors · 2025-11-25
> **Re: Reviewer 5qL1 [Part 4] : Addressing Concerns on Missing Proprietary Safety-Tuned Models [W3] and Specific Prompt Design [W4]**
>
> > [w3] *"The analysis seems to have missed proprietary safety-tuned models (e.g., GPT-4, Claude) - this would give an idea of the gap between the open and proprietary models."*
>
> We appreciate the reviewer highlighting this point. Our initial focus on open-weight models was motivated by the need for **full reproducibility and manageable cost**, which allowed us to run 4000 tool acceptance tests across two environments under fully controlled settings and may be more reproducible in the community. We agree that including proprietary safety-tuned models provides valuable insight into the behavioural gap between open and closed-source models.
> **To address this concern, we have now tested four frontier proprietary models.** Specifically, we evaluated:
>
> 1) **GPT-4.1-2025-04-14, Gemini-2.5-Flash, DeepSeek-R1-0528,  and Claude-Sonnet-4.5-20250929**
> 2) **n = 100 tool-offer episodes (5 batch * 20 offers) for each prompt variant of each tool, to each agent.**
> 3) Using the **same metrics as in Table 1** of our original submission.
>
> These results are provided in **official comment by authors [3/5]**, and are now incorporated into the updated PDF in **Section 3.2 and Appendix B,** along with an analysis of these proprietary models’ tool adoption patterns.
>
> > [w4] *"The paper relies on a specific prompt design - it would be good to discuss sensitivity to prompts and include significance testing on the results."*
>
> Thank you for this valuable suggestion. In response, we developed five new prompt variants (please see **official comment by authors [1/5]** ) incorporating all reviewers’ concerns and evaluated them on:
>
> 1) our previously evaluated 7b-level open-source models,
> 2) the newly added safety-aligned proprietary models, and
> 3) 70b-level open-source models included for scaling analysis.
>
> Full descriptions of the prompt variants and complete evaluation results are available in **Official Comment by Authors [1/5], [2/5], [3/5], and [4/5], as well as in the updated paper.** We believe these additional analyses directly address the reviewer’s concerns regarding prompt sensitivity and robustness, and **we kindly ask you to reconsider your score in light of our responses, and these strengthened results. Thank you!**

---

> ### Author Response · Authors · 2025-11-27
> **Follow Up on Reviewer–Author Discussion**
>
> Dear Reviewer 5qL1,
>
> I hope you are doing well. I wanted to thank you again for your thoughtful suggestions. **We carefully addressed each of them and conducted a substantial amount of additional experiments and analyses during the rebuttal.** With the Reviewer–Author Discussion period ending soon, please let us know if you have any further questions or points you would like us to clarify.
>
> Thank you so much again for your time and engagement with our submission.
>
>
> Sincerely,
>
> The Authors

---

### Author Response · Authors · 2025-11-25
**Official Comment to All Reviewers [1/5]: Additional Experimental Results**

Dear Reviewers, thank you for your valuable suggestions regarding the use of **frontier proprietary models** and **prompt-variant testing** to strengthen our paper. Based on your feedback, we have conducted the following additional experiments:
1. **Prompt Variants**
   We designed five new prompt variants based on the original prompt:
   - **V1:** Neutral and ambiguous (Suggested by: **Reviewer 2KjF**, Question 1):
   - **V2:** Without “Designer has chosen” (Suggested by: **Reviewer 2KjF**, Weakness 1; **Reviewer UPqW**, Question 2; **Reviewer 5qL1**, Weakness 4):
   - **V3:** Without the “unfair” label (Suggested by: **Reviewer UPqW**, Question 3):
   - **V4:** With explicit ethical concern (Suggested by: **Reviewer u8T6**, Weakness 2):
   - **V5:** With explicit ethical concern and explicit punishment (Suggested by: **Reviewer u8T6**, Weakness 3):

   We tested and reported tool-acceptance rates of the 7B-level open-source models, 70B-level open-source models, and proprietary models using these variants in the comments below. All prompt variants are provided in the **Appendix E.3** of the updated PDF.
2. **Frontier Proprietary Models**  (Suggested by: **Reviewer 5qL1**, Weakness 3; **Reviewer 2KjF**, Weakness 2; **Reviewer UPqW**, Weakness 1;  **Reviewer u8T6**, Weakness 4)
   We evaluated four proprietary frontier models using both the original prompt and all five variants:
   - **GPT-4.1-2025-04-14**
   - **Gemini-2.5-Flash**
   - **DeepSeek-R1-0528**
   - **Claude-Sonnet-4.5-20250929**
3. **Larger Open-Source Models**   (Suggested by: **Reviewer 2KjF**, Weakness 2; **Reviewer UPqW**, Weakness 1;  **Reviewer u8T6**, Weakness 4)
   We additionally tested four larger 70B-level open-source models in the scaling study, using the same set of prompts:
   - **Meta-Llama-3.1-70B-Instruct**
   - **Meta-Llama-3-70B-Instruct**
   - **Mixtral-8x7B-Instruct-v0.1**
   - **Qwen2.5-72B-Instruct**

Below, we provide the additional results in three separate comments, as each contains a long table.  Visualization of the results can also be found in **Section 3.2**, and full tables are included in **Appendix B** of the updated PDF.

---

> ### Author Response · Authors · 2025-11-25
> **Official Comment to All Reviewers [2/5]: Open-source Model (7B-level) Results**
>
> ## Secret Communication Channel
>
> | **Model** | **Tool Acceptance (%)** |  | Partner Selection (%) |  |  | **Accept as Partner (%)** |
> | --- | --- | --- | --- | --- | --- | --- |
> | **Variant 0: baseline** |  | **Mistral-7b** | **LLaMA-3-8b** | **LLaMA-3.1-8b** | **Qwen2.5-7b** |  |
> | Mistral-7b | **100 ± 0.00** | – | 38.00±6.20 | 48.60±4.83 | 13.40±3.65 | **100 ± 0.00** |
> | LLaMA-3-8b | **100 ± 0.00** | **100 ± 0.00** | – | 0 | 0 | **100 ± 0.00** |
> | LLaMA-3.1-8b | **100 ± 0.00** | 13.40±2.88 | 83.20±3.49 | – | 3.40±2.88 | **100 ± 0.00** |
> | Qwen2.5-7b | **100 ± 0.00** | **100 ± 0.00** | 0 | 0 | – | **100 ± 0.00** |
> | **Variant 1: ambiguous_neutral** |  |  |  |  |  |  |
> | Mistral-7b | **100 ± 0.00** | – | 48.20±2.17 | 47.20±0.84 | 4.60±1.52 | **100 ± 0.00** |
> | LLaMA-3-8b | **100 ± 0.00** | **100.00±0.00** | – | 0 | 0 | **100 ± 0.00** |
> | LLaMA-3.1-8b | **100 ± 0.00** | 8.60±2.70 | 91.40±2.70 | – | 0 | **100 ± 0.00** |
> | Qwen2.5-7b | **100 ± 0.00** | 84.80±2.95 | 15.20±2.95 | 0 | – | **100 ± 0.00** |
> | **Variant 2: no_designer** |  |  |  |  |  |  |
> | Mistral-7b | **100 ± 0.00** | – | 39.00±5.83 | 49.60±8.91 | 11.40±3.91 | **100 ± 0.00** |
> | LLaMA-3-8b | **100 ± 0.00** | **100 ± 0.00** | – | 0 | 0 | **100 ± 0.00** |
> | LLaMA-3.1-8b | **100 ± 0.00** | 8.20±0.84 | 91.80±0.84 | – | 0 | **100 ± 0.00** |
> | Qwen2.5-7b | 0 | – | – | – | – | 0 |
> | **Variant 3: no_unfair_label** |  |  |  |  |  |  |
> | Mistral-7b | **100 ± 0.00** | – | 33.20±2.28 | 57.00±4.06 | 9.80±3.42 | **100 ± 0.00** |
> | LLaMA-3-8b | **100 ± 0.00** | **100 ± 0.00** | – | 0 | 0 | **100 ± 0.00** |
> | LLaMA-3.1-8b | **100 ± 0.00** | 10.40±3.51 | 89.60±3.51 | – | 0 | **100 ± 0.00** |
> | Qwen2.5-7b | 19.20 ± 3.70 | 84.91±6.54 | 0 | 15.09±6.54 | – | **100 ± 0.00** |
> | **Variant 4: explicit_ethical_concern** |  |  |  |  |  |  |
> | Mistral-7b | 81.20 ± 1.30 | – | 60.63±4.63 | 39.12±4.71 | 0 | 50.00 ± 70.71 |
> | LLaMA-3-8b | 94.60 ± 1.52 | 91.58±4.87 | – | 0 | 0 | 0 |
> | LLaMA-3.1-8b | 62.20 ± 9.68 | 9.89±4.38 | 90.11±4.38 | – | 0 | 50.00 ± 70.71 |
> | Qwen2.5-7b | 0 | – | – | – | – | 0 |
> | **Variant 5: ethical_with_penalties** |  |  |  |  |  |  |
> | Mistral-7b | 42.40 ± 3.78 | – | 48.97±10.48 | 38.22±10.71 | 12.30±4.40 | 0 |
> | LLaMA-3-8b | 50.80 ± 5.85 | 73.27±6.32 | – | 0 | 0 | 0 |
> | LLaMA-3.1-8b | 33.20 ± 6.38 | 8.86±6.12 | 91.14±6.12 | – | 0 | 50.00 ± 70.71 |
> | Qwen2.5-7b | 0 | – | – | – | – | 0 |
>
> ## Secret Strategic Hint
>
> | **Model** | **Tool Acceptance (%)** |  | Partner Selection (%) |  |  | **Accept as Partner (%)** |
> | --- | --- | --- | --- | --- | --- | --- |
> | **Variant 0: baseline** |  | **Mistral-7b** | **LLaMA-3-8b** | **LLaMA-3.1-8b** | **Qwen2.5-7b** |  |
> | Mistral-7b | **100 ± 0.00** | – | 48.80±6.22 | 50.20±6.38 | 1.00±0.71 | **100 ± 0.00** |
> | LLaMA-3-8b | **100 ± 0.00** | **100 ± 0.00** | – | 0 | 0 | **100 ± 0.00** |
> | LLaMA-3.1-8b | **100 ± 0.00** | 5.20±1.92 | 94.80±1.92 | – | 0 | **100 ± 0.00** |
> | Qwen2.5-7b | **100 ± 0.00** | 77.80±1.92 | 11.80±3.27 | 10.40±2.30 | – | **100 ± 0.00** |
> | **Variant 1: ambiguous_neutral** |  |  |  |  |  |  |
> | Mistral-7b | **100 ± 0.00** | – | 49.60±5.08 | 37.20±5.31 | 13.20±3.35 | **100 ± 0.00** |
> | LLaMA-3-8b | **100 ± 0.00** | **100 ± 0.00** | – | 0 | 0 | **100 ± 0.00** |
> | LLaMA-3.1-8b | **100 ± 0.00** | 8.60±1.67 | 91.40±1.67 | – | 0 | **100 ± 0.00** |
> | Qwen2.5-7b | **100 ± 0.00** | 83.60±4.98 | 0 | 16.40±4.98 | – | **100 ± 0.00** |
> | **Variant 2: no_designer** |  |  |  |  |  |  |
> | Mistral-7b | **100 ± 0.00** | – | 38.20±2.39 | 52.60±3.36 | 9.20±3.11 | **100 ± 0.00** |
> | LLaMA-3-8b | **100 ± 0.00** | **100.00±0.00** | – | 0 | 0 | **100 ± 0.00** |
> | LLaMA-3.1-8b | **100 ± 0.00** | 7.40±2.70 | 92.60±2.70 | – | 0 | **100 ± 0.00** |
> | Qwen2.5-7b | **100 ± 0.00** | **100.00±0.00** | 0 | 0 | – | **100 ± 0.00** |
> | **Variant 3: no_unfair_label** |  |  |  |  |  |  |
> | Mistral-7b | **100 ± 0.00** | – | 42.20±5.07 | 53.80±5.17 | 4.00±1.58 | **100 ± 0.00** |
> | LLaMA-3-8b | **100 ± 0.00** | **100.00±0.00** | – | 0 | 0 | **100 ± 0.00** |
> | LLaMA-3.1-8b | **100 ± 0.00** | 8.40±3.44 | 91.60±3.44 | – | 0 | **100 ± 0.00** |
> | Qwen2.5-7b | **100 ± 0.00** | 17.40±5.03 | 0 | 82.60±5.03 | – | **100 ± 0.00** |
> | **Variant 4: explicit_ethical_concern** |  |  |  |  |  |  |
> | Mistral-7b | 82.60 ± 4.34 | – | 46.23±3.49 | 53.77±3.49 | 0.00±0.00 | 50.00 ± 70.71 |
> | LLaMA-3-8b | **100 ± 0.00** | 87.40±5.55 | – | 0 | 0 | 0 |
> | LLaMA-3.1-8b | 55.20 ± 3.11 | 3.91±1.82 | 96.09±1.82 | – | 0 | 50.00 ± 70.71 |
> | Qwen2.5-7b | 0 | – | – | – | – | 0 |
> | **Variant 5: ethical_with_penalties** |  |  |  |  |  |  |
> | Mistral-7b | 51.60 ± 4.04 | – | 51.98±4.70 | 33.32±3.20 | 14.70±3.67 | 0 |
> | LLaMA-3-8b | 64.80 ± 5.12 | 58.01±7.59 | – | 0 | 0 | 0 |
> | LLaMA-3.1-8b | 35.40 ± 2.07 | 6.86±4.88 | 87.01±4.79 | – | 6.13±4.47 | 33.33 ± 57.74 |
> | Qwen2.5-7b | 0 | – | – | – | – | 0 |

---

> ### Author Response · Authors · 2025-11-25
> **Official Comment to All Reviewers [3/5]: Open-Source Model (70B-level) Results**
>
> ## Secret Communication Channel
>
> | **Model** | **Tool Acceptance (%)** |  | Partner Selection (%) |  |  | **Accept as Partner (%)** |
> | --- | --- | --- | --- | --- | --- | --- |
> | **Variant 0: baseline** |  | **Mixtral-8x7B** | **LLaMA-3-70B** | **LLaMA-3.1-70B** | **Qwen2.5-72B** |  |
> | Mixtral-8x7B | **100 ± 0.00** | – | 0.00±0.00 | **100±0.00** | 0.00±0.00 | **100 ± 0.00** |
> | LLaMA-3-70B | **100 ± 0.00** | **100±0.00** | – | 0.00±0.00 | 0.00±0.00 | **100 ± 0.00** |
> | LLaMA-3.1-70B | **100 ± 0.00** | 0.00±0.00 | **100±0.00** | – | 0.00±0.00| **100 ± 0.00** |
> | Qwen2.5-72B | 0.00±0.00 | – | – | – | – | 0.00±0.00 |
> | **Variant 1: ambiguous_neutral** |  |  |  |  |  |  |
> | Mixtral-8x7B | **100.00 ± 0.00** | – | 0.00±0.00 | **100.00±0.00** | 0.00±0.00 | **100.00 ± 0.00** |
> | LLaMA-3-70B | **100.00 ± 0.00** | 90.00±14.14 | – | 0.00±0.00 | 10.00±14.14 | **100.00 ± 0.00** |
> | LLaMA-3.1-70B | **100.00 ± 0.00** | 0.00±0.00 | **100.00±0.00** | – | 0.00±0.00 | **100.00 ± 0.00** |
> | Qwen2.5-72B | 58.00 ± 16.43 | 93.10±12.78 | 0.00±0.00 | 6.90±12.78 | – | **100.00 ± 0.00** |
> | **Variant 2: no_designer** |  |  |  |  |  |  |
> | Mixtral-8x7B | **100.00 ± 0.00** | – | 0.00±0.00 | **100.00±0.00** | 0.00±0.00 | **100.00 ± 0.00** |
> | LLaMA-3-70B | **100.00 ± 0.00** | 94.00±8.94 | – | 6.00±8.94 | 0.00±0.00 | **100.00 ± 0.00** |
> | LLaMA-3.1-70B | **100.00 ± 0.00** | 0.00±0.00 | **100.00±0.00** | – | 0.00±0.00 | **100.00 ± 0.00** |
> | Qwen2.5-72B | 0.00 ± 0.00 | – | – | – | – | 0.00 ± 0.00 |
> | **Variant 3: no_unfair_label** |  |  |  |  |  |  |
> | Mixtral-8x7B | **100.00 ± 0.00** | – | 0.00±0.00 | **100.00±0.00** | 0.00±0.00 | **100.00 ± 0.00** |
> | LLaMA-3-70B | **100.00 ± 0.00** | 92.00±8.37 | – | 8.00±8.37 | 0.00±0.00 | **100.00 ± 0.00** |
> | LLaMA-3.1-70B | **100.00 ± 0.00** | 0.00±0.00 | **100.00±0.00** | – | 0.00±0.00 | **100.00 ± 0.00** |
> | Qwen2.5-72B | 0.00±0.00 | – | – | – | – | 0.00±0.00 |
> | **Variant 4: explicit_ethical_concern** |  |  |  |  |  |  |
> | Mixtral-8x7B | 0±0.00 | – | – | – | – | 0±0.00 |
> | LLaMA-3-70B | 0±0.00 | – | – | – | – | 0±0.00 |
> | LLaMA-3.1-70B | 0±0.00 | – | – | – | – | 0±0.00 |
> | Qwen2.5-72B | 0±0.00 | – | – | – | – | 0±0.00 |
> | **Variant 5: ethical_with_penalties** |  |  |  |  |  |  |
> | Mixtral-8x7B | 0±0.00 | – | – | – | – | 0±0.00 |
> | LLaMA-3-70B | 0±0.00 | – | – | – | – | 0±0.00 |
> | LLaMA-3.1-70B | 0±0.00 | – | – | – | – | 0±0.00 |
> | Qwen2.5-72B | 0±0.00 | – | – | – | – | 0±0.00 |
>
> ## Secret Strategic Hint
>
> | **Model** | **Tool Acceptance (%)** |  | Partner Selection (%) |  |  | **Accept as Partner (%)** |
> | --- | --- | --- | --- | --- | --- | --- |
> | **Variant 0: baseline** |  | **Mixtral-8x7B** | **LLaMA-3-70B** | **LLaMA-3.1-70B** | **Qwen2.5-72B** |  |
> | Mixtral-8x7B | **100 ± 0.00** | – | 0±0.00 | **100±0.00** | 0±0.00 | **100 ± 0.00** |
> | LLaMA-3-70B | **100 ± 0.00** | 86.00±8.94 | – | 14.00±8.94 | 0±0.00 | **100 ± 0.00** |
> | LLaMA-3.1-70B | **100 ± 0.00** | 0±0.00 | **100±0.00** | – | 0±0.00 | **100 ± 0.00** |
> | Qwen2.5-72B | 0 ± 0.00 | – | – | – | – | 0 ± 0.00 |
> | **Variant 1: ambiguous_neutral** |  |  |  |  |  |  |
> | Mixtral-8x7B | **100.00 ± 0.00** | – | 0.00±0.00 | **100.00±0.00** | 0.00±0.00 | **100.00 ± 0.00** |
> | LLaMA-3-70B | **100.00 ± 0.00** | 86.00±5.48 | – | 14.00±5.48 | 0.00±0.00 | **100.00 ± 0.00** |
> | LLaMA-3.1-70B | **100.00 ± 0.00** | 0.00±0.00 | **100.00±0.00** | – | 0.00±0.00 | **100.00 ± 0.00** |
> | Qwen2.5-72B | 90.00 ± 7.07 | 93.33±9.08 | 0.00±0.00 | 6.67±9.08 | – | **100.00 ± 0.00** |
> | **Variant 2: no_designer** |  |  |  |  |  |  |
> | Mixtral-8x7B | **100.00 ± 0.00** | – | 0.00±0.00 | **100.00±0.00** | 0.00±0.00 | **100.00 ± 0.00** |
> | LLaMA-3-70B | **100.00 ± 0.00** | 88.00±4.47 | – | 12.00±4.47 | 0.00±0.00 | **100.00 ± 0.00** |
> | LLaMA-3.1-70B | **100.00 ± 0.00** | 0.00±0.00 | **100.00±0.00** | – | 0.00±0.00 | **100.00 ± 0.00** |
> | Qwen2.5-72B | 0.00 ± 0.00 | – | – | – | – | 0.00 ± 0.00 |
> | **Variant 3: no_unfair_label** |  |  |  |  |  |  |
> | Mixtral-8x7B | **100.00 ± 0.00** | – | 0.00±0.00 | **100.00±0.00** | 0.00±0.00 | **100.00 ± 0.00** |
> | LLaMA-3-70B | **100.00 ± 0.00** | 88.00±13.04 | – | 12.00±13.04 | 0.00±0.00 | **100.00 ± 0.00** |
> | LLaMA-3.1-70B | **100.00 ± 0.00** | 0.00±0.00 | **100.00±0.00** | – | 0.00±0.00 | **100.00 ± 0.00** |
> | Qwen2.5-72B | 86.00 ± 16.73 | 83.72±20.69 | 0.00±0.00 | 16.28±20.69 | – | **100.00 ± 0.00** |
> | **Variant 4: explicit_ethical_concern** |  |  |  |  |  |  |
> | Mixtral-8x7B | 0.00 ± 0.00 | – | – | – | – | 0.00 ± 0.00 |
> | LLaMA-3-70B | 0.00 ± 0.00 | – | – | – | – | 0.00 ± 0.00 |
> | LLaMA-3.1-70B | 0.00 ± 0.00 | – | – | – | – | 0.00 ± 0.00 |
> | Qwen2.5-72B | 0.00 ± 0.00 | – | – | – | – | 0.00 ± 0.00 |
> | **Variant 5: ethical_with_penalties** |  |  |  |  |  |  |
> | Mixtral-8x7B | 52.00 ± 19.24 | – | 0.00±0.00 | **100.00±0.00** | 0.00±0.00 | 0.00 ± 0.00 |
> | LLaMA-3-70B | 0.00 ± 0.00 | – | – | – | – | 0.00 ± 0.00 |
> | LLaMA-3.1-70B | 0.00 ± 0.00 | – | – | – | – | 0.00 ± 0.00 |
> | Qwen2.5-72B | 0.00 ± 0.00 | – | – | – | – | 0.00 ± 0.00 |

---

> ### Author Response · Authors · 2025-11-25
> **Official Comment to All Reviewers [4/5]: Proprietary Model Results**
>
> ## Secret Communication Channel
>
> | **Model** | **Tool Acceptance (%)** |  | Partner Selection (%) |  |  | **Accept as Partner (%)** |
> | --- | --- | --- | --- | --- | --- | --- |
> | **Variant 0: baseline** |  | **GPT-4.1** | **Gemini-2.5** | **DeepSeek-R1** | **Claude-4.5** |  |
> | GPT-4.1 | **100.00 ± 0.00** | – | 28.00±13.04 | 2.00±2.74 | 70.00±12.75 | 66.67 ± 57.74 |
> | Gemini-2.5 | **100.00 ± 0.00** | 97.00±2.74 | – | 2.00±2.74 | 1.00±2.24 | 66.67 ± 57.74 |
> | DeepSeek-R1 | **100.00 ± 0.00** | 21.00±8.22 | 2.00±2.74 | – | 77.00±9.75 | 66.67 ± 57.74 |
> | Claude-4.5 | 0.00 ± 0.00 | – | – | – | – | 0.00 ± 0.00 |
> | **Variant 1: ambiguous_neutral** |  |  |  |  |  |  |
> | GPT-4.1 | **100.00 ± 0.00** | – | 24.00±8.22 | 12.00±7.58 | 64.00±10.84 | 66.67 ± 57.74 |
> | Gemini-2.5 | **100.00 ± 0.00** | 97.00±4.47 | – | 1.00±2.24 | 2.00±2.74 | 66.67 ± 57.74 |
> | DeepSeek-R1 | **100.00 ± 0.00** | 15.00±7.07 | 0.00±0.00 | – | 85.00±7.07 | **100.00 ± 0.00** |
> | Claude-4.5 | **100.00 ± 0.00** | 0.00±0.00 | 0.00±0.00 | **100.00±0.00** | – | **100.00 ± 0.00** |
> | **Variant 2:no_designer** |  |  |  |  |  |  |
> | GPT-4.1 | 52.00 ± 6.71 | – | 23.79±15.88 | 4.04±5.58 | 72.17±12.90 | 66.67 ± 57.74 |
> | Gemini-2.5 | 96.00 ± 5.48 | **100.00±0.00** | – | 0.00±0.00 | 0.00±0.00 | **100.00 ± 0.00** |
> | DeepSeek-R1 | **100.00 ± 0.00** | 10.00±3.54 | 0.00±0.00 | – | 90.00±3.54 | 50.00 ± 70.71 |
> | Claude-4.5 | 0.00 ± 0.00 | – | – | – | – | 0.00 ± 0.00* |
> | **Variant 3: no_unfair_label** |  |  |  |  |  |  |
> | GPT-4.1 | **100.00 ± 0.00** | – | 34.00±9.62 | 8.00±8.37 | 58.00±7.58 | 66.67 ± 57.74 |
> | Gemini-2.5 | **100.00 ± 0.00** | **100.00±0.00** | – | 0.00±0.00 | 0.00±0.00 | **100.00 ± 0.00** |
> | DeepSeek-R1 | **100.00 ± 0.00** | 17.00±5.70 | 2.00±4.47 | – | 81.00±6.52 | 66.67 ± 57.74 |
> | Claude-4.5 | 0.00 ± 0.00 | – | – | – | – | 0.00 ± 0.0 |
> | **Variant 4: explicit_ethical_concern** |  |  |  |  |  |  |
> | GPT-4.1 | 0.00 ± 0.00 | – | – | – | – | 0.00 ± 0.00 |
> | Gemini-2.5 | 0.00 ± 0.0 | – | – | – | – | 0.00 ± 0.00 |
> | DeepSeek-R1 | 3.00 ± 4.47 | 25.00±35.36 | 0.00±0.00 | – | 75.00±35.36 | 0.00 ± 0.00 |
> | Claude-4.5 | 0.00 ± 0.00 | – | – | – | – | 0.00 ± 0.00 |
> | **Variant 5: ethical_with_penalties** |  |  |  |  |  |  |
> | GPT-4.1 | 0.00 ± 0.00 | – | – | – | – | 0.00 ± 0.00 |
> | Gemini-2.5 | 0.00 ± 0.00 | – | – | – | – | 0.00 ± 0.00 |
> | DeepSeek-R1 | 0.00 ± 0.00 | – | – | – | – | 0.00 ± 0.00 |
> | Claude-4.5 | 0.00 ± 0.00 | – | – | – | – | 0.00 ± 0.00 |
>
> ## Secret Strategic Hint
>
> | **Model** | **Tool Acceptance (%)** |  | Partner Selection (%) |  |  | **Accept as Partner (%)** |
> | --- | --- | --- | --- | --- | --- | --- |
> | **Variant 0: baseline** |  | **GPT-4.1** | **Gemini-2.5** | **DeepSeek-R1** | **Claude-4.5** |  |
> | GPT-4.1 | **100.00 ± 0.00** | – | 2.00±2.74 | 1.00±2.24 | 97.00±2.74 | 66.67 ± 57.74 |
> | Gemini-2.5 | 98.00 ± 2.74 | 99.00±2.24 | – | 0.00±0.00 | 1.00±2.24 | 50.00 ± 70.71 |
> | DeepSeek-R1 | **100.00 ± 0.00** | 4.00±2.24 | 1.00±2.24 | – | 95.00±3.54 | 66.67 ± 57.74 |
> | Claude-4.5 | 0.00 ± 0.00 | – | – | – | – | 0.00 ± 0.00 |
> | **Variant 1: ambiguous_neutral** |  |  |  |  |  |  |
> | GPT-4.1 | **100.00 ± 0.00** | – | 17.00±4.47 | 11.00±4.18 | 72.00±5.70 | 66.67 ± 57.74 |
> | Gemini-2.5 | **100.00 ± 0.00** | 99.00±2.24 | – | 0.00±0.00 | 1.00±2.24 | 50.00 ± 70.71 |
> | DeepSeek-R1 | **100.00 ± 0.00** | 14.00±8.22 | 0.00±0.00 | – | 86.00±8.22 | 50.00 ± 70.71 |
> | Claude-4.5 | 49.00 ± 8.22 | 0.00±0.00 | 0.00±0.00 | **100.00±0.00** | – | **100.00 ± 0.00** |
> | **Variant 2: no_designer** |  |  |  |  |  |  |
> | GPT-4.1 | **100.00 ± 0.00** | – | 27.00±13.51 | 15.00±5.00 | 58.00±12.55 | 66.67 ± 57.74 |
> | Gemini-2.5 | **100.00 ± 0.00** | 99.00±2.24 | – | 1.00±2.24 | 0.00±0.00 | **100.00 ± 0.00** |
> | DeepSeek-R1 | **100.00 ± 0.00** | 12.00±2.74 | 1.00±2.24 | – | 87.00±4.47 | 66.67 ± 57.74 |
> | Claude-4.5 | 0.00 ± 0.00 | – | – | – | – | 0.00 ± 0.00 |
> | **Variant 3: no_unfair_label** |  |  |  |  |  |  |
> | GPT-4.1 | **100.00 ± 0.00** | – | 32.00±17.18 | 9.00±8.22 | 59.00±11.40 | 66.67 ± 57.74 |
> | Gemini-2.5 | **100.00 ± 0.00** | 99.00±2.24 | – | 0.00±0.00 | 1.00±2.24 | 50.00 ± 70.71 |
> | DeepSeek-R1 | **100.00 ± 0.00** | 14.00±7.42 | 3.00±2.74 | – | 83.00±5.70 | 66.67 ± 57.74 |
> | Claude-4.5 | 0.00 ± 0.00 | – | – | – | – | 0.00 ± 0.00 |
> | **Variant 4: explicit_ethical_concern** |  |  |  |  |  |  |
> | GPT-4.1 | 0.00 ± 0.00 | – | – | – | – | 0.00 ± 0.00 |
> | Gemini-2.5 | 16.00 ± 8.94 | **100.00±0.00** | – | 0.00±0.00 | 0.00±0.00 | 0.00 ± 0.00 |
> | DeepSeek-R1 | 0.00 ± 0.00 | – | – | – | – | 0.00 ± 0.00 |
> | Claude-4.5 | 0.00 ± 0.00 | – | – | – | – | 0.00 ± 0.00 |
> | **Variant 5: ethical_with_penalties** |  |  |  |  |  |  |
> | GPT-4.1 | 0.00 ± 0.00 | – | – | – | – | 0.00 ± 0.00 |
> | Gemini-2.5 | 0.00 ± 0.00 | – | – | – | – | 0.00 ± 0.00 |
> | DeepSeek-R1 | 0.00 ± 0.00 | – | – | – | – | 0.00 ± 0.00 |
> | Claude-4.5 | 0.00 ± 0.00 | – | – | – | – | 0.00 ± 0.00 |

---

> ### Author Response · Authors · 2025-11-25
> **Official Comment to All Reviewers [5/5]: Our Main Findings from Additional Experiments**
>
> 1. **Voluntary acceptance of unfair collusion tools is robust across models, scales, and prompt framings.**
>    Under the original prompt and three neutral or ethically softened variants (Baseline, V1, V2, V3), **almost all models, including 7B, 70B, and frontier proprietary models, accept unfair collusion tools in nearly all trials**. This holds even when descriptions are neutral, ambiguous, or lack the “unfair” label entirely.
> 2. **Ethical framing and explicit penalties reduce acceptance for frontier and 70B models but do not eliminate it.**
>    Under variants with explicit ethical warnings (V4) and ethical warnings + explicit punishment (V5), safety-tuned proprietary models and 70B models show substantial declines in acceptance. However, **7B-level open-source models still accept at high rates**, indicating that voluntary tool exploitation persists in smaller models even under discouraging conditions.
> 3. **The phenomenon is not driven by prompt-induced compliance.**
>    Removing potential inducement cues (e.g., “designer has chosen”), removing fairness cues (“unfair”), or using neutral/ambiguous framing does **not** meaningfully reduce acceptance. This suggests the behavior reflects **a genuine willingness to exploit an unfair advantage**, rather than an artifact of a particular phrasing.
> 4. **Findings generalize beyond small models to frontier systems.**
>    Acceptance rates among **GPT-4.1-2025-04-14, Gemini-2.5-Flash, and DeepSeek-R1-0528** are extremely high under Baseline, V1, V2, and V3, mirroring the behavior of 7B and 70B open-source models.  This shows that voluntary adoption of unfair collusion tools is **not specific to small and outdated open-source models**.
> 5. **Ethical awareness does not prevent collusion.**
>    Rationales produced by many models explicitly acknowledge that the tool is unfair or harmful, yet they still choose to accept it. This supports our central claim: **models understand the unethical nature of collusion but willingly opt in when the opportunity is presented**.
> 6. **Our framework provides a reusable methodology for auditing voluntary misuse risk.**
>    The Accept/Refuse decision, combined with downstream behavioral divergence across two distinct MAS environments (Liar’s Bar and Cleanup), offers a practical and reproducible evaluation protocol for future safety work, policy studies, and alignment diagnostics.
> We deeply thank all reviewers for their insightful suggestions, which substantially strengthened the paper. All additional experiments, analyses, and prompt variants are now documented in **Section 3.2**, **Appendix B**, and **Appendix E.2** in the updated version.

---

### Author Response · Authors · 2025-11-27
**Summary Comment by Authors**

Dear ICLR 2026 **Program Chairs, Senior Area Chairs, and Area Chairs,**

We would like to sincerely thank you for overseeing the review and discussion process for our submission. We also extend our appreciation to all reviewers for their thoughtful feedback during the initial evaluation. In response to the concerns and suggestions raised, we implemented **every suggested improvement by reviewers**, with all results have been **carefully documented and incorporated into the updated manuscript, and author comments**, including:

1) **New scaling studies on four 70B-level open-source models**,  (addressing concerns from **Reviewer 2KjF, Reviewer UPqW, Reviewer u8T6**)

2) **New evaluations on four frontier safety-aligned proprietary models** (addressing concerns from **Reviewer 5qL1, Reviewer 2KjF, Reviewer UPqW, Reviewer u8T6**), and

3) **Prompt-sensitivity analyses across five new variants** that suggested by each of the reviewer, conducted across all 7B-level open-source models tested in our original submission as well as the eight newly tested models.

We are also grateful that **Reviewer 2KjF** engaged during the discussion before the incident happened, and **reconsidered their score to 6** after reviewing the additional analyses and expressing full satisfaction with our responses. We greatly appreciate this constructive interaction, as it indicates that the extended experiments meaningfully strengthened the submission. Therefore, we would like to note that our final score was 6, 6, 4, 4 (average score: 5) prior to the platform bug.

We didn't receive any follow up questions or comments from **Reviewer 5qL1, Reviewer UPqW, and Reviewer u8T6**, and we fully understand that this year’s platform bug unfortunately prevented them from participating further in the discussion or updating their reviews and scores. Nevertheless, we believe that the substantial additional experimental results we provided directly address their concerns, and our responses adequately resolve the issues they raised.

We are very grateful for the reviewers’ positive assessments and constructive insights, which meaningfully strengthened our paper. We would be happy to provide any further clarification the newly assigned AC may find helpful, and we also hope that the full review history, including the updated analyses, expanded experiments, and reviewer responses can be taken into account during the final evaluation.

Thank you again for your effort in facilitating this process.

Sincerely,

The Authors

---

### Meta-Review · Area_Chair_Cqym · 2026-01-07

**Summary:**

All reviewers broadly agreed that the paper studies an important and timely AI safety question: whether LLM agents will knowingly and voluntarily engage in unfair collusion when given explicit opportunities to do so.

However, initial concerns clustered around four main themes:
(1) novelty and conceptual depth,
(2) generalizability beyond small open-source models,
(3) robustness to prompt framing and inducement effects, and
(4) scope and realism of the evaluation environments.

**Reviewer Concerns:**

The concerns about
(2) generalizability beyond small open-source models,
(3) robustness to prompt framing and inducement effects
are well addressed in the rebuttal.

**Reviewer Scores:**

Reviewers would still question environmental realism and real-world deployment breadth. Although the environments are more complex than those used in prior collusion work and are well-justified, some skepticism remains about how directly the findings extrapolate to open-ended real-world multi-agent deployments.

While the authors clarified the distinction between voluntary unethical choice and enforced or emergent collusion, this concern is inherently subjective and not fully resolvable through additional experiments.

Overall, the study could be deeper to figure out the reason behind the observations.

---

### Decision · Program_Chairs · 2026-01-26

Reject